# Assessing the Effects of Cancer Diagnosis and Coping Strategies on Patients in Vhembe District Hospitals, Limpopo Province

**DOI:** 10.3390/nursrep15070222

**Published:** 2025-06-20

**Authors:** Dorah Ursula Ramathuba, Takalani Friddah Rafundisani, Maria Sonto Maputle

**Affiliations:** Department of Advanced Nursing Science, University of Venda, Thohoyandou 0950, South Africa; rafundisani9@gmail.com (T.F.R.); sonto.maputle@univen.ac.za (M.S.M.)

**Keywords:** cancer diagnosis, coping, illness perception, oncology

## Abstract

**Background**: Unlike other chronic diseases, cancer patients undergo different types of treatments that affect their well-being, and as a result, they tend to have different experiences from those of other chronic disease sufferers. The purpose of this study was to assess the effects of cancer diagnosis and coping strategies on patients in Vhembe District hospitals in Limpopo Province. **Methodology**: The study design used was a quantitative descriptive cross-sectional survey. The target population included patients in the Vhembe District of Limpopo who had started treatment within the last year. Probability-stratified sampling was used to sample 207 patients from seven selected hospitals in Vhembe District. A self-administered questionnaire was used to collect data, and the data were analyzed using a software package for descriptive statistics (SPSS version 23). Tables were used to display the results visually, and chi-square tests were used to compare the variables. Ethical principles were considered for the participants’ privacy, anonymity, and informed consent. **Findings**: The findings revealed that the majority of patients 185 (89.4%) experienced a sense of psychosocial distress such as emotional pain; 142 (68.6%) participants experienced hopelessness and despair, 127 (61.3%) resorted to substance use, 160 (77.3%) did not have a positive attitude towards seeking the medical and other support resources available, only a minority resorted to spirituality, and 121 (63.2%) indicated seeking further clarity about the disease. The study recommends supporting cancer patients and their families through the cancer journey. **Contributions**: Clinicians should provide psychosocial support interventions to enhance mental health and quality of life in cancer patients, and decentralize oncology services by including primary care professionals in delivering chronic illness disease management strategies.

## 1. Introduction

The World Health Organization [1] indicates that over 35 million new cancer cases are predicted by 2050. The rapidly growing global cancer burden is associated with socioeconomic development and is mostly felt in low- and middle-income countries. In South Africa, cancer is the fifth leading cause of death, with an estimated 33,800 premature deaths annually, contributing to a 0.8-year reduction in average population life expectancy. The most prevalent cancers are reproductive and lung cancers. Prostate cancer is a significant concern, accounting for 13% of male deaths, and with over 4000 new cases diagnosed annually among women, the lifetime risk of breast cancer is 1 in 27. Other common cancers include lung, bronchus, colorectal, and cervical cancer [1].

South Africa has a two-tier health care system: a public system managed by the Department of Health and a private system operated by private providers and funded through private insurance and out-of-pocket payments. This situation contributes to poor and inequitable access to cancer services. Cancer services and resources are placed in provincial and tertiary hospitals, with the rural peripheral districts left underdeveloped. Chitha et al. [2] also indicate similar sentiments that resources are more likely to be found in urban areas, tertiary centers, and quaternary hospitals, without taking the socioeconomic, value, and cultural contexts into account when designing cancer care pathways.

Many South African people continue to experience inequities in accessing cancer care, even though the National Cancer Strategic Framework 2017–2022 assures that cancer prevention strategies are enhanced through the tobacco control program, including the inclusion of Hepatitis B and HPV vaccination in the extended immunization program [3]. The national cancer care strategy is still not well coordinated, cancer care pathways are not structured, and infrastructure and equipment are still lacking for diagnosis and the initiation of treatment. Furthermore, palliative and psychotherapeutic care are limited, if not missing, especially in rural areas such as Vhembe where the primary health care facilities have limited social and psychological support services as most hospitals lack psychologists.

Other factors are related to sociocultural factors. Some rural communities may have cultural beliefs or practices that discourage seeking help for mental health issues related to cancer diagnoses, contributing to poor coping mechanisms, social isolation, and other stressors associated with the adverse effects of treatment. A cancer diagnosis has different impacts on different patients and causes them to develop perceptions, attitudes, and beliefs about cancer; these eventually determine how the patients behave afterward. Nipp et al. [4] posit that patients with incurable cancer often endure numerous physical and emotional challenges and must decide if the benefits of treatment outweigh the significant toxicities and unwanted side effects of cancer therapy, which influence their ability to make decisions regarding both their cancer care and their care at the end of their life.

Rural areas like Vhembe are geographically isolated, with low levels of mental literacy, making it difficult for patients to connect with social networks and form support groups. Khalfi et al. [5] indicate that cancer patients develop affective disorders within a year of diagnosis, and most of the time, their coping mechanisms can fail, meaning that they require psychiatric management. Furthermore, Ownby [6] also notes that distress is an unpleasant experience of an emotional, psychological, social, or spiritual nature, and that only a small number of distressed patients are identified and treated, which is the case in rural communities as their cultural beliefs or practices discourage seeking help for mental health issues, leading to delays in seeking psychotherapy.

Patient navigation is a cornerstone of patient-centered care in high-income settings. While patient navigation programs are progressively gaining traction in LMICs, not all settings have instituted such programs [7]. Rural health services in Limpopo Province lack psychotherapy services, which impacts the quality of life of cancer patients, and it is an area that has been neglected. Cancer causes psychological distress due to its incurable nature and treatment side effects, resulting in stress, anxiety, and depression. Similarly, rural patients may have fewer opportunities for social support from friends, family, or community members, making it harder to cope with stressors, as their cancer may be a new phenomenon. Patient navigators are exceptionally trained to provide holistic support, encompassing practical, emotional, and psychosocial assistance, and bridge the gap between patients and available resources by facilitating access to financial aid and social support programs [8]. If this navigation model can be adopted in the public health system and with government support, patients might experience less distress. Being diagnosed with cancer is one of the most feared phenomena among human beings because it negatively impacts patients in many ways [4]. Cancer treatment and coping strategies usually depend on the impact of the disease on the perceptions, attitudes, and beliefs about the disease that patients have, as these play a crucial role in the patient’s level of understanding of the disease [9]. Cruz & Echeverría [10] indicate that 62% of patients used denial as a coping strategy, and 60% did the same with cognitive avoidance, which suggests that these are strategies frequently used by most cancer patients. Doherty et al. [11] indicate that coping is a stage of recurrent change through which individuals face the demands of their sociocultural contexts that determine the reciprocal relationship between acting internally and externally. Thus, this study wanted to understand how a cancer diagnosis affects quality of life and which coping strategies are used. Only when adequate social support is provided can patients make informed decisions. The stressors that preoccupy cancer patients are trauma, anxiety, and the loss of the will to live, as well as other mental disharmonies that serve as underlying threats to patients’ quality of life and well-being, resulting in a loss of sense of self, poor decision-making regarding treatment, and a decrease in quality of life [6]. With well-coordinated cancer care support, patients will be able to access psychotherapy and adopt positive coping strategies that will improve their quality of life. Access to decentralized, equitable cancer services by region may contribute to the quality of life of cancer patients. Thus, this study aimed to assess the effects of cancer diagnosis and coping strategies on patients in Vhembe District hospitals in Limpopo Province.

## 2. Methodology

### 2.1. Research Design

A quantitative approach was used as it provided a high level of measurement and reliability. The study adopted a cross-sectional survey strategy with a quantitative design to measure the prevalence of cancer health outcomes, understand how cancer diagnoses affects quality of life and coping strategies, and describe the features of the population, utilizing a questionnaire for data collection. The design assisted in describing the respondents’ perceptions of how cancer diagnosis had affected their lives and coping strategies in the Vhembe District of Limpopo Province.

### 2.2. Setting

This study was conducted in Vhembe District, one of the six regions of the Limpopo Province. The district has seven hospitals, one regional hospital and six district hospitals. Cancer patients consult at these hospitals and are referred to a Provincial Hospital for further cancer treatment management since district hospitals lack oncology services. The district hospitals have no specialist oncology teams, only general practitioners and a few oncology-trained nurses.

#### Population and Sampling

According to Creswell & Creswell [12], a study usually utilizes a population, a group of individuals or events with the same characteristics from which a sample will be selected. Our population consisted of all patients diagnosed with cancer; our target population was those already on treatment for a year. A stratified sampling method was used to calculate a representative sample of respondents per district hospital, ensuring that everyone in the population had an equal probability of being selected. However, during data collection, a convenience sample was used to access those patients who were available and consented on the particular day the researcher visited the hospital, ensuring that the sample would be representative of the population and produce more statistically valid results that could be generalized to describe or predict the characteristics of the whole population [12].

### 2.3. Data Collection

Surveys use structured questionnaires as collection techniques because they typically involve collecting data on many variables from a large and representative sample of respondents [13]. Data collection in this study used the questionnaire method to elicit respondents’ answers on the effects of cancer diagnosis and coping strategies post-cancer diagnosis. The questionnaire was self-developed from the literature in consultation with a statistician and was then translated into local languages. The questionnaire had four sections: section one covered the demographic profile, and section two covered the prevalence of cancer, aspects of cancer diagnosis, time taken to await results and diagnosis, time taken for the initiation of treatment, treatment duration, and types of treatments. The third section focused on the effects of cancer diagnosis and treatment, patients’ feelings and reactions to their cancer diagnosis, and the physical, social, financial, and emotional impacts of cancer diagnosis. Section four covered coping strategies and mechanisms. During data collection, the researcher was present to check the completeness of the questionnaire, ensuring that there were no questions missed or spoiled questionnaires.

### 2.4. Rigor and Data Collection Instrument

Data was collected using a pre-tested and structured questionnaire. The content and face validity of the questionnaire in this study were determined by an expert statistician and the supervisor, who assisted in correcting ambiguous, complex, and obscure questions. Furthermore, ineffective and irrelevant questions were discarded from the questionnaire. To cater for reliability, Pearson’s product was used to assess the instrument’s reliability, the correlation coefficient was used, and a value of 0.80 was obtained, which indicated a good relationship [14]. A pilot study was conducted in one of the district hospitals on a sample of twenty patients to test the tool’s effectiveness. On observation, most respondents preferred the English version of the questionnaire, and the questionnaire was completed in twenty-five minutes. All completed questionnaires were checked for completeness and consistency.

### 2.5. Data Analysis

The process of data analysis is used to generate operational and beneficial information from the accumulated results of a study, and quantitative data analysis techniques provide researchers with tools to draw inferences on perceptions, beliefs, and knowledge about a subject of discussion, as well as statistically proving a change or variation in knowledge structures of the population. In this study, SPSS version 23.0 was used to process and analyze data quantitatively. The data were coded so that both descriptive and analytic statistics could be used to analyze the results obtained. The cross-tabulation of demographic data and other variables against perceptions was performed to obtain the associations and trends. The results of the data analysis were presented in tables.

### 2.6. Ethical Considerations

Ethical clearance (SHS/17/PDC/18/1207) was provided by the University Research Ethics Committee (UREC), and approval was provided by the provincial health department. According to Sivasubramaniam et al. [15], research ethics should be taken as moral values required to guide the researcher in carrying out research using the most appropriate research methodology while being responsible and morally considerate of others. This research study considered several ethical issues, such as agreements from individual respondents, anonymity and privacy for respondents, the protection of respondents and other researchers from harm, and avoiding deception, as Sivasubramaniam et al. [15] suggest. The researcher explained to respondents the purpose of the study, the procedures involved, the potential risks/benefits, how confidentiality would be maintained, and the right to withdraw from participation. The researcher ensured that the explanation was at the respondents’ comprehension or understanding level. Informed consent was obtained when the respondents signed the forms agreeing to complete the questionnaires.

## 3. Findings

A total of 207 patients diagnosed with cancer participated (Table 1); 129 (62.3%) were females and 78 (37.7%) were males. The age range was between 25 and 58 years, and the mean age for females was 44 vs. 40 years old for males. Most respondents, 104 (50%), were unemployed and depended on child support grants; only a minority, 95 (45.9%), received salaries from informal employment. The majority, 174 (54%), had a secondary school education.

Incidence of cancer

Cancer prevalence was higher among respondents above the age of 30 years, accounting for 190 (91.8%) of the respondents. The most common cancer types among female respondents were breast cancer, at 57 (27.5%), and cervical cancer, at 52 (25.1%), while among male respondents, the common types of cancer were prostrate, at 33 (15.9%), lung cancer, at 19 (9.2%), and colon, with 16 cases (7.7%).

Modes of treatment

Respondents received various treatment modalities, with the number of patients receiving chemotherapy alone being 91 (44.0%); 44 (21.2%) respondents were treated with radiation and 72 (34.8%) respondents were treated using both types.

Effects of illness perceptions and psychological distress

The majority of respondents, 127 (61.3%), confirmed that they resorted to alcohol or drugs because it calmed them. Eighty-nine percent, 185 (89.4%), of the cancer patients insisted on being left alone to deal with their cancer diagnosis. Most of the respondents, 143 (69.5%), spent many sleepless nights pondering over their cancer diagnosis. Again, 142 (68.6%) respondents affirmed cancer as their fate and were no longer prepared to fight it. A total of 160 (77.3%) had developed wrong perceptions that no one knew how to care for them, expressing anger and fear. One hundred and twenty-one (63.2%) of the respondents confirmed that cancer forced them to look for more information whenever they received some bad news. One hundred and one (51.6%) responded that they were trying to lighten up and see the humor in challenging situations.

The chi-square test was used, based on cross-tabulating items for each research question instead of a hypothesis, and only significant results at *p* < 0.05 were reported. A correlation was made to determine the effects of cancer diagnosis and the coping strategies used by respondents, and the findings revealed that there was an association between receiving a cancer diagnosis and self-consoling by singing religious or emotive songs as a coping strategy thought to solve one’s illness problems when left alone, with a *p*-value of 0.02.

Respondents who wished to be left alone were more likely to avoid joining a cancer support group for counseling, as shown by a significant *p*-value association of *p* = 0.10, were less likely to focus on the positive aspects of their lives while waiting for a treatment plan, with a *p*-value = 0.33, and were also less likely to consult medical staff on their treatment plan, *p* = 0.01.

An association of *p* = 0.29 also existed between the consumption of drugs and alcohol to ease the burden of cancer diagnosis, the belief that the cancer burden can be relieved by getting drunk, and the use of intoxicating substances. Again, a significant relationship existed, at *p* = 0.418, with patients choosing a coping strategy that diverted the patients’ attention from the illness to something else. The coping strategy of keeping quiet in public places, *p* = 0.44, was significantly associated with the desire to be left alone to deal with one’s problems.

## 4. Discussion

Cancer, in the nature its of existence, is a traumatic event that causes stressful experiences, resulting in the development of the most severe psychological depressive symptoms. Cancer diagnosis represents a psychological trauma for the patient. Regardless of any physical suffering, it is accompanied by several unwanted experiences that hugely disturb individuals’ well-being and quality of life. A cancer diagnosis brings about shock and anger in most patients. Thus, they perceive the illness differently, and their attitude toward life determines the treatment outcomes or prognosis of the disease. The cancer patients in the study experienced physical, financial, emotional, and psychosocial stress. A wide range of collaborative, supportive care is required from multiple disciplines in order to deliver highly specialized care, such as assisting patients in coping with the effects of their disease and treatment.

The majority of cancer patients, 185 (89.4%), insisted on being left alone to deal with their cancer diagnosis. To be left alone is to try to find answers, and some of them indicated that they resorted to singing spiritual songs to ease the emotional distress. According to Cinar et al. [16], spirituality is defined as the meaning and purpose of life, and psychological and spiritual coping methods are preferred mostly by women, older people, and those at lower socioeconomic levels. Various authors indicate that the patients who experienced intense spiritual emotions turned to religion, which was found to be the most frequent coping strategy in cancer patients, and that religious attitudes had the most significant impact on coping with stress [16,17].

Again, being alone is a sign of withdrawal or self-isolation, which may lead to depression and other maladaptive behaviors such as drowning one’s sorrows, as 127 (61.3%) of the respondents resorted to alcohol or drug use. Jones et al. [18] agree that substance use disorders, such as alcohol and drugs, are prevalent at higher rates among survivors of certain types of cancers. Our findings indicate an association between the consumption of drugs and alcohol to ease the burden of cancer diagnosis, *p* < 0.29, which are negative coping mechanisms detrimental to health and disease outcomes. Yusufov et al. [19] indicate that individuals with cancer may use substances in attempts to cope with psychological distress or poorly controlled physical symptoms, sometimes referred to as “chemical coping”, which may lead to comorbidities that negatively impact cancer care. Furthermore, substance use disorders are associated with demographic variables such as race and age, which relate to study findings that poverty and socioeconomic conditions are contributing factors [20]. South Africa is among the top five countries in the world for alcohol consumption. This high consumption rate is driven by a culture of drinking to intoxication, especially on weekends, and a high prevalence of binge drinking.

A diagnosis of cancer invariably brings thoughts of mortality to the forefront of patients’ minds. It may be associated with social isolation and personal stressors such as a lack of social support, living below the poverty line, and poor access to oncology care, as experienced by respondents in the study, which may lead to suicidal ideation and is also shown by the significant association *p* < 0.10 that they were unlikely to seek social support through methods such as joining support groups. Isolation and lack of family support may lead to isolation and suicidal thoughts [21,22]. Cancer diagnosis is associated with a higher probability of suicidal ideation; the prevalence of suicidal ideation in other studies varies widely, ranging from 0.7 to 46.3% (Kolva et al. [23]; China was 24.95% [24]). Chen et al. [22] postulate that being unmarried, less educated, living alone, living in rural areas, having a lower quality of social functioning, and having financial problems are linked to suicidal ideation, which was valid for most of the respondents in the study, who were financially unstable and lacking social support.

Cancer diagnosis is a life stressor and can affect the psychological and physical state of patients, and respondents in the study indicated a lack of sleep. Al Maqbali [25] suggests that more than half of cancer patients experience sleep disturbance, which negatively affects the quality of life among patients diagnosed with cancer and may result in poor healing, decreased cognitive functioning, and reduced work activity. Zeng et al. [26,27] posit that females have a higher prevalence of sleep disturbance than males since women are less likely to receive social support. Rural women are caregivers who must take care of their families and households while enduring the effects of a cancer diagnosis, which results in psychological stress. Similarly, most respondents in the study were unemployed, depended on the childcare social grant, lacked social support, and suffered from anxiety and the anticipation of treatment outcomes. The uncoordinated cancer care service contributed to their uncertainty about the effects of treatment, fear of disease progression and death, and hopelessness, leading to sleep anxiety. Sleep-related problems could be reflections of anxious symptomatology, with subsequent repercussions during nocturnal rest periods. Clinicians must assist and plan adequate palliative treatments to improve the quality of life and promote the well-being of patients with cancer [28].

Cancer patients usually lose hope due to treatment failure and other symptomatic effects that impact their physical and psychological outlooks and end up demoralizing patients. The majority of the respondents, 142 (68.6%), affirmed that they had lost hope and were no longer prepared to fight the disease. Li et al. [29] also suggest that patients with higher levels of hope had lower burdens of symptoms from cancer. Kunitake et al. [30] indicated in their study that patients with enterostomy experienced different of psycho-emotional conditions, decreased levels of hope, and increased levels of anxiety and depression, which relates to this study’s findings where some participants resorted to isolating themselves, abusing alcohol, or focusing on spirituality. Nural et al. [31] affirm that hopelessness observed in patients who stated that they have no fear of death may be associated with their loss of faith in staying alive and their belief that they will never again return to their previous state of health. They have abandoned the battle against death. Nikoloudi et al. [32] indicate that hope is crucial for effective adaptation to the disease and a strategy for coping with physical and mental distress, as it enhances disease management, psychological adaptation, and overall quality of life. Palliative care plays a crucial role in managing hopelessness in cancer patients by addressing physical, emotional, and spiritual distress, promoting hope-boosting interventions, and providing personalized support to improve quality of life.

Cancer diagnosis not only affects patients but also their families and caregivers who always feel depressed, anxious, and afraid. These effects are common and normal responses to this life-changing experience. This study assessed the effects of cancer diagnosis and coping strategies, and the findings revealed that cancer patients in rural communities experience a psychosocial and emotional burden of disease due to the inaccessibility and inequalities of cancer care services.

The results indicate that cancer is prevalent among rural communities; 129 (62.3%) females and 78 (37.7%) males were affected by different types of cancers. The prevalence of breast cancer was 57 (27.5%) and cervical cancer was 52 (25.1%), respectively. For males, prostate cancer’s prevalence was 33 (15.9%), followed by lung cancer at 19 (9.2%) and colon cancer at 16 (7.7%). Bray et al. [33] indicate that the most prevalent carcinoma is lung cancer, which is responsible for almost 2.5 million new cases, or one in eight cancers, worldwide (12.4% of all cancers globally), followed by cancers of the female breast (11.6%), colorectum (9.6%), prostate (7.3%), and stomach (4.9%). Similarly, in South Africa, the most prevalent male cancers are prostate, colorectal, and lung cancer, whilst among females the most comment cancers are breast, cervix, and colorectal cancers. These results are consistent with the literature; in South Africa, cervical cancer incidence and mortality rates are higher among black African women; this is likely due to late diagnosis and the high burden of HIV in this population group. For white females, the highest incidence is breast cancer, while for males, prostate cancer rates are higher among white men than black men. This suggests that the incidence rates reflect diagnosis at more advanced stages of the disease, possibly due to poor access to cancer screening, treatment, and inadequate knowledge of the signs and symptoms of different types of cancers. The South African health system’s cancer care services are not in rural areas but in urban metropolitan areas [34]. Chitha et al. [2] in their study undertaken in Mpumalanga and the Western Cape, also indicate similar sentiments that resources are more likely to be found in urban areas, tertiary centers, and quaternary hospitals. District health services only provide primary health care, and health education and awareness are limited as nurses are overworked and hospitals understaffed.

The mean age of respondents in the study was 50 years old; most respondents were diagnosed in their late thirties, for females, or late forties, for males. According to StatsSA [34], the median age at diagnosis of cancer was 59 years for females and 64 years for males in 2018. However, the median age at death due to cancer was 62 for females and 64 for males, suggesting that cancer in males may be diagnosed at more advanced stages than in females. Thompson et al. [35] state that women seek more health care in response to physical and mental health concerns due to their unique reproductive health care needs, and often report longer consultation times than men. In rural communities, men use traditional common-knowledge herbal treatments to cleanse their systems since herbal medicine is freely available and accessible without consultation with traditional healers. Ozioma & Chinwe [36] indicate that at least 80% of Africans still rely on medicinal plants for health care, and these medicines have gained momentum due to their low cost, affordability, availability, acceptability, and low toxicity. Reddy et al. [37] concur that other sociocultural factors still prevent rural communities from benefiting from quality health services, such as using multiple sources, including traditional healers, before they approach expert medical care, which results in late presentation and the late initiation of treatment [38].

The two-tier health system contributes to inaccessibility of cancer care as patients cannot afford private health care and have to wait for a prolonged period to be diagnosed and for treatment to be initiated due to under-resourced public health institutions; even at tertiary public health services, oncologists are limited, and by the time patients are seen the disease will have progressed. Mncedisi [39] indicates that the scarcity of oncology specialists disproportionately impacts urban and rural communities across low-income African countries, which lack trained oncologists, with median ratios of clinical oncologists that are 0.006 per 100,000 people and 0.01 oncologists per 100 cancer patients. This poses a critical challenge in addressing the global burden of cancer.

Furthermore, the urban metropolitan in Gauteng faces the highest burden of cancer care in the country as the province’s central hospitals were not originally designed to handle the current load since their oncology services are extended beyond the province through referrals and also across national borders, offering highly specialized care that is difficult to replicate at tertiary and district hospitals, which often lack essential surgical, diagnostic, and pathological capabilities. There is a lack of proper coordination and a structure of oncology services. Chita et al. [2] also concur that poorly developed care pathways currently limit South Africa’s cancer care services, including the lack of standardized budgets within and between provinces, the fact that cancer registries lack resources, and the poor implementation of existing skills and programs.

Community-centered care is practiced in primary health care facilities, and when patients are discharged back to their homes in communities, cancer care services should continue. However, there are limited oncology nurses and poor coordination and support for cancer patients in rural villages. Hui et al. [40] indicate that oncology nurses provide psychoeducation on symptom management, emotional support, information, coaching, and linkage of resources via home visits and telephone support. In rural communities, patients die in isolation without this support, as there are no oncology nurse navigators for support.

Psychosocial support is crucial in cancer care, and practical and open communication is essential in identifying the needs of patients. Social support involves the willing or actual provision of relationships, information, advice, or assistance that enables a person to successfully cope with the day-to-day challenges of crisis in their personal life. A total of 160 (77.3%) respondents had wrong perceptions that no one knew how to care for them, expressing anger and fear. Barnes et al. [41] suggest that rural people with cancer and those who care for them experience different health care services by the very nature of their rurality as compared with people from urban areas because of the underutilization of offered services due to both practical barriers (e.g., distance, expense, and time) and intrapersonal barriers.

Furthermore, Corovic et al. [42] purport that system-related barriers include the limited availability of palliative care services, poor communication between teams, a lack of interdisciplinary communication, and low insurance reimbursement. Additionally, most health care providers felt that care could improve by having the same team see patients in inpatient and outpatient settings for better continuity between inpatient and outpatient care and better communication [43]. Interdisciplinary teams are needed to deliver multidimensional care. Oncology teams are essential in providing supportive care on the front lines and referring patients to supportive care services such as palliative care, social work, rehabilitation, psycho-oncology, and integrative medicine [44].

One hundred and twenty-one (63.2%) of the respondents confirmed their fear of death. The diagnosis of a terminal illness, such as cancer, can cause fear, anxiety, and stress [44,45]. Furthermore, Soleimani et al. [46] indicate that death anxiety and psychological distress are more significant among females than males, both in general and among cancer patients. Rodenbach et al. [47] suggest that worrying about dying is a barrier to patients’ acceptance of their terminal illness and can have significant harmful effects, including worsening physical and psychological symptoms and overall quality of life.

This study’s findings indicate that respondents found meaning in spirituality by distracting themselves from negative thoughts through singing spiritual songs. Spiritual experiences can bring comfort, meaning, and peace, and give one a sense that life is complete. Lim et al. [48] indicate that coping strategies that are actively confronting, such as expressing feelings and asking others for support, and meaning-based, like positive reframing and spirituality, may better facilitate death acceptance, and that it is a strong predictor and promoter of psychological health. Furthermore, Xing et al. [49] also indicate that spirituality can increase resistance against mental health crises following the diagnosis and treatment of cancer in patients.

## 5. Conclusions

Poor and uncoordinated oncology care, insufficient oncology resources, and limited health care infrastructure intensify disparities in cancer care, hindering the effectiveness of treatment, having a direct impact on the quality of patient care, and contributing to poorer health outcomes and increased mortality rates among cancer patients. A conceptual model of cancer care encompassing different aspects of the cancer care continuum, from prevention and early detection to diagnosis, treatment, survivorship, and end-of-life care, is required for rural communities to access quality care to achieve a good quality of life.

## 6. Limitations of the Study

Since this study is cross-sectional, cause-and-effect impacts of the disease cannot be measured at one snapshot of time. There was a limitation in the use of convenience sampling and recruiting more females during data collection, which may have resulted in biased findings and the reduced generalizability of the study.

## Figures and Tables

**Table 1 nursrep-15-00222-t001:** Psychological distress effects and coping strategies used by patients to deal with cancer diagnosis.

n = 207								
Illness Perception/Psychological Distress	Coping Strategy	Observed			Expected	Chi-Square	Df	*p*-Value
		Yes	No	Total				
Alcohol helps calm me down when I am upset.	Using drugs, alcohol, or something to forget the ordeal/thoughts of cancer diagnosis							
	Agree	80	39	119	69.0	2.899	2	0.029
	Neutral	10	5	15	69.0			
	Disagree	40	33	73	69.0			
	Diverting my attention to something else							
	Agree	89	21	110	69.0	1.052	2	0.0418
	Neutral	21	9	30	69.0			
	Disagree	51	16	67	69.0			
I wish people would leave me alone to face my problems.	Using drugs or alcohol to forget the ordeal/thoughts of a cancer diagnosis							
	Agree	76	39	115	69.0	7.926	2	0.019
	Disagree	34	40	74	69.0			
	Neutral	9	9	18	69.0			
	Joining a cancer support group for counseling							
	Agree	34	81	115	69.0	9.293	2	0.010
	Neutral	8	10	18	69.0			
	Disagree	38	36	74	69.0			
	Focusing on the positive aspects of life while waiting for a treatment plan							
	Agree	40	75	115	69.0	7.830	2	0.020
	Neutral	8	10	18	69.0			
	Disagree	41	33	74	69.0			
	Talking to medical staff about a treatment plan							
	Agree	43	72	115	69.0	13.579	2	0.001
	Neutral	12	6	18	69.0			
	Disagree	46	28	74	69.0			
	Singing religious or emotive songs to comfort myself							
	Agree	89	26	115	69.0	12.491	2	0.002
	Neutral	8	10	18	69.0			
	Disagree	43	31	74	69.0			
	Keeping quiet when in public places							
	Agree	93	22	115	69.0	6.239	2	0.044
	Neutral	10	8	18	69.0			
	Disagree	53	21	74	69.0			
No matter what I do I cannot sleep in fear of death.	Focusing on the positive aspects of life while waiting for a treatment plan							
	Agree	36	55	91	69.0	6.842	2	0.033
	Neutral	21	12	33	69.0			
	Disagree	32	51	83	69.0			
	Reading books on cancer							
	Agree	31	66	97	69.0	8.692	2	0.013
	Neutral	21	6	27	69.0			
	Disagree	35	48	83	69.0			
I think cancer is my fate, and there is no point in fighting it.	Using drugs, alcohol, or something to forget the ordeal/thoughts of a cancer diagnosis							
	Agree	49	39	88	69.0	6.877	2	0.032
	Neutral	25	7	32	69.0			
	Disagree	45	42	87	69.0			
	Avoiding any topic about cancer							
	Agree	67	21	88	69.0	7.487	2	0.024
	Neutral	31	1	32	69.0			
	Disagree	65	22	87	69.0			
Having cancer is bad enough. But to make matters worse, no one knows how to care for me.	Using drug. Alcohol, or something to forget the ordeal/thoughts of cancer diagnosis t							
	Agree	54	55	109	69.0	7.701	2	0.021

## Data Availability

All data supporting this manuscript have been made available. All data presented and analyzed during this study are included in this article.

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
