# Peer review of "Assessing the Effects of Cancer Diagnosis and Coping Strategies on Patients in Vhembe District Hospitals, Limpopo Province"

_nursrep, 2025, doi:10.3390/nursrep15070222_

Round 1

Reviewer 1 Report

Comments and Suggestions for Authors

Dear authors,

This study addresses an important and timely topic with potential contributions to culturally sensitive psycho-oncology research. The inclusion of culturally specific coping mechanisms is valuable and adds novelty to the literature.

However, the manuscript requires several critical revisions to improve its clarity, scientific rigor, and organization:

  1. The manuscript contains vague terminology that show be clarified for improved readability and scientific tone. Multiple sections-including the discussion-lack logical flow and coherent structure. Ensure that paragraphs remain focused and logically develop the topic introduced in the opening sentences.
  2. A clear research hypothesis or specific objectives is missing and should be added. The stated goal of assessing the “effects” of coping strategies implies causation, which is not supported by the cross-sectional descriptive design. Please reframe the study’s aims to align with the capabilities of the methodology.
  3. The manuscript does not sufficiently define the specific knowledge gap it addresses. Please elaborate on the study’s scientific contribution to the field, especially regarding coping strategies within underrepresented or culturally distinct populations.
  4. Provide more detail on how the questionnaires were developed or validated and how completeness and consistency checks were performed. In statistical analysis, the manuscript relies exclusively on chi-square tests. Further information is needed about the hypotheses tested, how expected values were calculated, how missing data were handled, and whether assumptions were verified.
  5. Results are currently presented only in table format. Consider using appropriate visualizations to enhance clarity and reader engagement.
  6. There’s an area in the discussion that shifts topics without clear connection to the data. Please revise to ensure internal coherence. 

Comments on the Quality of English Language

I recommend a professional Enghish language editor with experience in academic writing to improve sentence structure and overall flow. 

Author Response

The comments are uploaded, I am struggling with pasting

Reviewer 2 Report

Comments and Suggestions for Authors

The proposed manuscript entitled “Assessing the effects of cancer diagnosis and coping strategies of patients in Vhembe district hospitals, Limpopo Province-main” investigates the psychosocial impact of a cancer diagnosis and the coping mechanisms employed by patients in the Vhembe District of Limpopo Province, South Africa. The study utilizes a quantitative descriptive cross-sectional survey design, collecting data from 207 cancer patients via a self-administered questionnaire. The authors aim to shed light on the challenges faced by these patients and propose recommendations for improved support and healthcare delivery. Although the manuscript could be of some interest, there are significant areas of improvement that need to be addressed, before considering the manuscript for publication. Here is a point-by-point analysis divided in major and minor areas of improvement.

Major Areas of Improvement:

  1. The methods section requires some improvements. While the text states a quantitative approach provides "a high level of measurement and reliability", a brief justification for why a cross-sectional survey was specifically chosen for this research question would strengthen this section. 
  2. The description of the population as “all cancer patients who had commenced or were already on treatment” can be considered good. However, it's stated the target population included patients "diagnosed with cancer in the past year" in the abstract. Clarify if these are the same or slightly different criteria.   
  3. “A simple random selection method was used”. While simple random sampling is ideal, in a hospital setting, was it truly random, or was it a convenience sample of available and willing patients who met the criteria? If the latter, this should be acknowledged as a potential limitation.
  4. It's mentioned the questionnaire was “pre-tested and structured” and “pilot-tested”. How? Briefly describe the pilot testing process (e.g., sample size, population used for pilot testing) and if any modifications were made based on the pilot results. This adds to the robustness of the methodology.
  5. The paragraph starting "Poverty has many negative impacts..." seems like a discussion point rather than a direct finding from the collected data, unless this was a specific question on the survey. Clarify if this is an interpretation or a direct finding.
  6. The presentation of chi-square results in the text (e.g., "X2 =11.997; n=207; Df=3; p<0.017)" ) is appropriate. However, ensure consistency in notation (e.g., "X2" vs. "X2"). Standard statistical notation should be used. 
  7. Some p-values are reported as "<0.017", "<0.033", "<0.002", "<0.010", "<0.020", "<0.001", "<0.029", "<0.019", "<0.044". It's generally preferred to report exact p-values unless they are very small (e.g., p < .001). If SPSS provides exact p-values like 0.017, 0.033, etc., it's better to report them as p = 0.017, p = 0.033. The use of "<" suggests the value is less than that threshold but not the exact value. Clarify this. For example, if p=0.017, it is less than 0.05, but reporting the exact value is more precise.   
  8. There seems to be some repetition in the paragraph discussing correlations. For instance, the finding about being left alone and avoiding support groups is mentioned twice with the same statistics. This should be consolidated. Similarly for focusing on positive aspects and consulting medical staff.   
  9. Table 3 and Table 4 present detailed chi-square results. The titles are "Illness perception and coping strategies used upon cancer diagnosis" and "Psychological distress effects and coping strategies used by patients to deal with cancer diagnosis". Ensure the distinction between these tables is clear and that items are not unnecessarily duplicated or could be combined if they represent similar constructs. The "Observed" and "Expected" columns in these tables are useful for chi-square interpretation but check if the "Expected" values are always necessary for the main manuscript table or could be detailed in supplementary materials if journal space is a concern.
  10. The limitation regarding the sample composition (patients diagnosed within a year and at different cancer stages) is also valid and well-noted. Please, consider briefly mentioning how this heterogeneity might have influenced the results if possible. 
  11. Please consider other factors, like diet, that could further influence habits that lead to cancer development, regardless of the geographical area. Please consider citing PMID 38474758 and further corroborate modifying factors that could lead to cancer development.

Minor Areas of Improvement:

  1. The abstract mentions "majority of patients160 (77.3%), experienced a sense of psychosocial distress, 123 (63.2%) emotional pain, and perceived loss of hope and despair and 185(89,4%) patients adopted adaptive and 127(61.3%) resorted to maladaptive coping strategies". This sentence is a bit convoluted and could be broken down for better readability. It's also unclear if the 89.4% adaptive and 61.3% maladaptive coping strategies are mutually exclusive or if patients could employ both. 
  2. The keyword "cancer patients" is redundant given "Cancer" and "oncology" are already listed. Please remove it and consider other options if needed.
  3. Authors should consider revising the manuscript in each section to ensure smoother transitions between paragraphs. Also, please ensure all statements derived from specific sources are appropriately and consistently cited. Nonetheless, check the manuscript for potential misplacing and errors of format. For instance, formatting of Table 1 and subsequent tables (e.g., inconsistent spacing, the "n=207" placement) needs to be standardized and made cleaner for publication. This is likely a formatting issue in the PDF provided.
  4. The methodology section is generally listed as a number. This heading should be titled "Methodology" or "Methods."
  5. The text describes percentages and numbers for various demographic categories (e.g., age, employment, income source). This is followed by Table 1. Ensure the narrative doesn't just repeat what's in the table but rather highlights key findings or trends from the table. For example, instead of just listing percentages, a sentence could summarize: "A significant portion of the patient cohort was over 50 years old and unemployed, with a higher proportion of females relying on grants".
  6. Please, revise the discussion to avoid starting multiple sentences with "The findings..." or similar structures, if possible, to improve flow.

Reviewer 3 Report

Comments and Suggestions for Authors

In this study, the authors provided a comprehensive overview of the psychological and coping challenges faced by cancer patients in the Vhembe district. Below are a number of issues that the authors shall address or revise:

  1. In this study, simple random sampling was used, which could not reflect the diversity of different cancer patients. It is better to apply a stratified sampling approach to these samples.
  2. In the discussion part, the findings on maladaptive coping strategies (e.g., alcohol use) are valuable, and the authors could further explore cultural or community-specific factors influencing these behaviors in the Vhembe context.
  3. Some formatting mistakes in the format should be revised:

1) The title of section 2 is needed.

2) Table 3 and Table 4, “Using drug. alcohol or something to forget the ordeal/thoughts of cancer diagnosis” and “Using drug. Alcohol, or something to forget the ordeal/thoughts of cancer diagnosis” should be “Using drug, alcohol or something to forget the ordeal/thoughts of cancer diagnosis”.

3) Table 4 lacked some lines at the end of the table.

Author Response

The paste options does not allow

Reviewer 4 Report

Comments and Suggestions for Authors

Dear Authors,

I would like to thank the authors for their effort in exploring the psychological responses and coping strategies of cancer patients in the Vhembe district of South Africa. The focus on the psychosocial impact of cancer diagnosis in a rural African context is timely and relevant, especially given the limited empirical data available in such settings. However, I believe the manuscript requires substantial revision in terms of structure, clarity, and scientific rigor in order to fully meet the standards of publication in Nursing Reports. Below, I outline specific concerns regarding each section of the manuscript.

Please refer to the attached file for details. I hope that my comments will contribute, even in a small way, to the improvement of your manuscript.

Best regards,

Author Response

I have uploaded the comments

Round 2

Reviewer 1 Report

Comments and Suggestions for Authors

Overall, your study addresses important and timely issues, and the core aims are presented clearly. To improve the manuscript, in the introduction, reorganize content from a broad to a more specific focus to improve coherence and readability. Correct the calculation errors in the findings to ensure that the numbers and percentages reported are accurate and consistent. In addition, in the discussion, maintain a logical progression from psychosocial impacts, to your study’s findings, to comparison with broader data, and then to health system factors-using transition sentences to clarify connections between these themes. These revisions will strengthen the clarity and impact of your paper. 

Comments on the Quality of English Language

The overall quality of English in the manuscript is generally adequate; however, it need some improvement. Many sentences are overly long and contain multiple ideas, which reduces clarity. Dividing these into shorter sentences would enhance readability. Additionally, there is a tendency to change topics within the same sentence or paragraph without appropriate transitions, making the text difficult to follow. There are also inconsistencies in the use of commas and periods, especially when presenting numerical data and percentages. For example, “185(89,4%)” should be written as “185 (89.4%),” with correct spacing and punctuation. Commas are sometimes incorrectly placed after percentage phrases, such as in “89.4%), experienced” and “77.3%),not.” These commas should be omitted unless the clause actually ends there.

Author Response

The comments have been uploaded as supplementary materials

Reviewer 2 Report

Comments and Suggestions for Authors

Authors addressed correctly the required revisions. 

Comments on the Quality of English Language

The manuscript probably needs some tweaking regarding English, though. 

Author Response

The manuscript has been revised, long sentences shortened, and punctuation corrected.

Reviewer 3 Report

Comments and Suggestions for Authors

I am satisfied with the author’s responses to my issues raised in my initial review. The revised manuscript is easier to follow based on feedback from the reviewers. I recommend that the revised paper be accepted.

Author Response

The manuscript have been revised and grammatical errors and punctuations improved. Long sentences shortened.

Reviewer 4 Report

Comments and Suggestions for Authors

Dear Authors,

Thank you for revising and adding content based on the reviewer comments within such a short period of time. I have confirmed that the comments have been appropriately addressed and reflected in the manuscript.

Best regards,

Author Response

The manuscript has been revised.